# Leveraging platinum-protein interactions to overcome chemoresistance

Fang Wang [1,2,3,7] ✉, Jonathan Braverman [2,4,7] ✉, George Eng[2,5], Ozen Leylek[2], Nicholas L. Petrone [3], Daniel S. Honeycutt [3], Shinya Imada[2], Brian Pallares [6], Daiyao Zhang [2], Jason M. Mrosla [6], Camellia S. Huang [2], Anna A. Griadunova[4], William K. McCarthy [6], Jacob M. Goldberg [6], Michael T. Hemann [2], Stephen J. Lippard[1] ✉ & Ömer H. Yilmaz [2,5] ✉

A common mechanism by which cancer cells acquire resistance to chemotherapeutics is through the overexpression of efflux pumps, enabling the removal of cytotoxic agents, such as anthracycline drugs. However, platinum anticancer agents that crosslink DNA and interact with proteins are poor efflux pump substrates. Here, we design dual warhead drug conjugates by tethering a platinum pharmacophore to the doxorubicin backbone. These drug conjugates retain the anticancer activity of anthracyclines and exhibit the ability to both circumvent drug efflux and delay the acquisition of drug resistance. In vivo experiments demonstrate that such drug conjugates extend survival in a preclinical organoid-based model of metastatic colon cancer in mice. Mechanistic studies indicate that these drug conjugates overcome resistance through covalent platinum-protein interactions, leading to significantly improved drug retention and alteration of subcellular drug distribution. This application of platinum offers many opportunities to confront issues related to chemoresistance and alternative pathways for augmenting conventional chemotherapeutics.

Chemotherapy is a critical component of modern cancer treatment. The development of chemoresistance, however, contributes to poor prognoses. Multidrug resistance (MDR)—the loss of response to structurally unrelated small-molecule anticancer drugs—often results in limited treatment options[1–4]. Extensive in vitro and clinical studies have revealed an association between MDR and the overexpression of several drug efflux transporters of the ATP-binding cassette (ABC) family, particularly the multidrug resistance protein 1 (MDR1, or P-glycoprotein, P-gp)[1,5–7]. The broad substrate scope and frequent overexpression of these transporters make them appealing therapeutic targets. Although P-gp inhibitors are effective in vitro,

they often lead to toxicities or pharmacokinetic changes when paired with chemotherapeutics in the clinic, thus offering limited utility[1,2,8–13]. These discouraging results are partly attributable to the protective role of P-gp in non-cancerous tissues to remove cytotoxins and xenobiotics. The nonselective blocking of ABC transporter activity can thus impede systemic drug clearance, resulting in side effects[1,9,14]. P-gp can be co-expressed in cancer cells alongside other ABC transporters, including multidrug resistance-associated protein 1 (MRP1) and the ABC subfamily G member 2 (ABCG2)[4,9,11,15]. To prevent drug efflux effectively, it may be necessary to inhibit multiple ABC transporters simultaneously, either with multiple

[1]Department of Chemistry, Massachusetts Institute of Technology, Cambridge, MA, USA. [2]Department of Biology, The David H. Koch Institute for Integrative Cancer Research at MIT, Massachusetts Institute of Technology, Cambridge, MA, USA. [3]Department of Chemistry, University of Rhode Island, Kingston, RI, USA. [4]Innovative Genomics Institute, University of California, Berkeley, Berkeley, CA, USA. [5]Department of Pathology, Beth Israel Deaconness Medical Center, Massachusetts General Hospital and Harvard Medical School, Boston, MA, USA. [6]Department of Chemistry, Colgate University, Hamilton, NY, USA. [7]These authors contributed equally: Fang Wang, Jonathan Braverman. ✉e-mail: fangwang@uri.edu; braverman@berkeley.edu; lippard@mit.edu; ohyilmaz@mit.edu

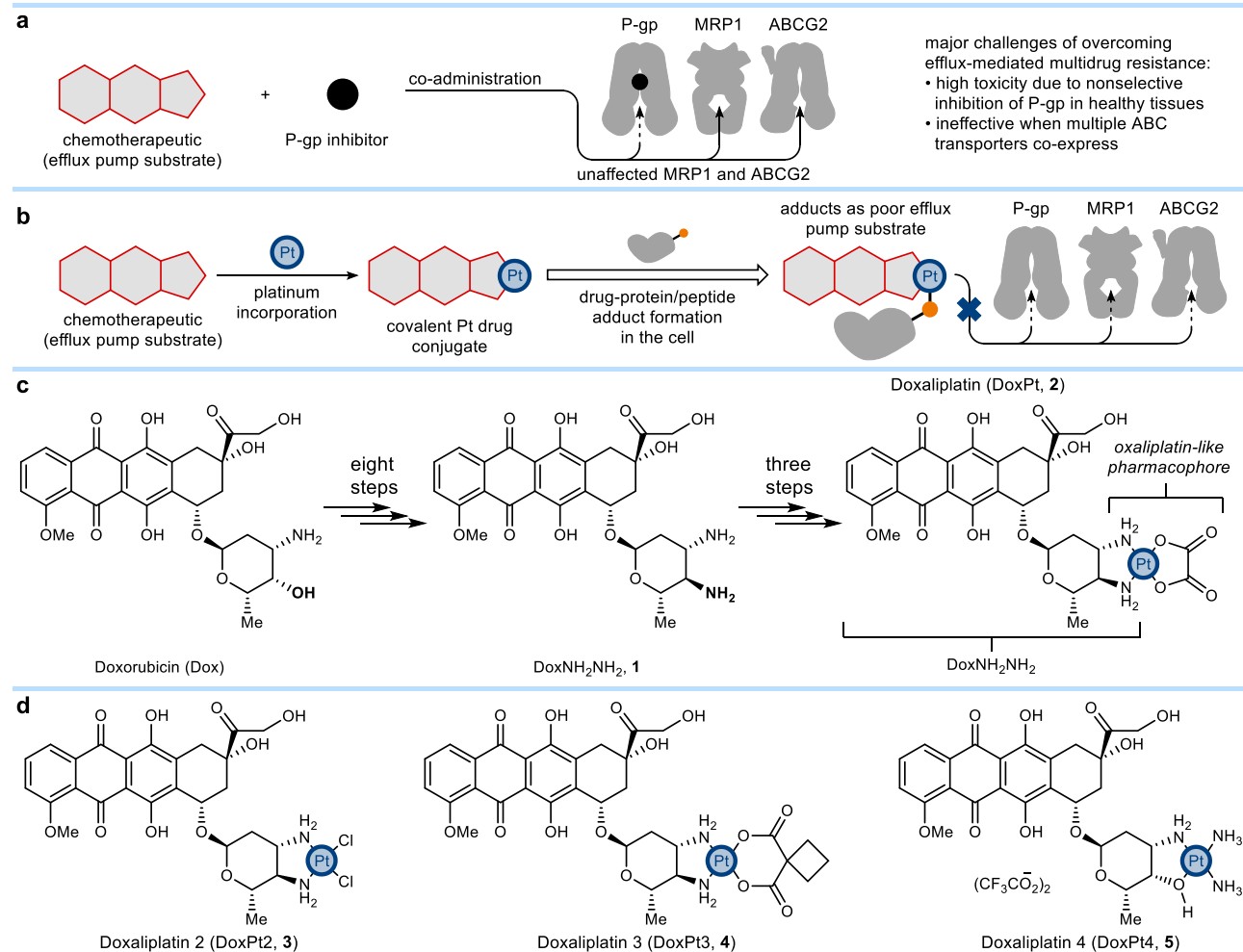

**Fig. 1 | Development of a strategy enabled by platinum-protein interactions to overcome chemoresistance. a** Key challenges associated with combination therapies using conventional ATP-binding cassette (ABC) transporter inhibitors to overcome multidrug resistance mediated by P-glycoprotein (P-gp), multidrug resistance protein 1 (MRP1), and ATP-binding cassette super-family G member 2 (ABCG2). **b** Platinum-containing drug conjugates exploit platinum-protein inter-actions to overcome efflux pump-mediated multidrug resistance without disrupt-ing normal efflux activity. **c** Synthesis of the doxorubicin-oxaliplatin conjugate, doxaliplatin (DoxPt, **2**), with minimal structural modifications to both parental drugs. **d** Doxaliplatin derivatives with different structural features.

inhibitors or less specific inhibitors, further complicating treatment regimens (Fig. 1a).

Here, we report the development of a strategy that leverages underexplored protein-platinum interactions to circumvent MDR without disrupting ABC transporter activity. A series of platinum-containing drug conjugates is generated by synthetically tethering anthracyclines to platinum moieties, including those derived from clinically used oxaliplatin, cisplatin, and carboplatin. These anticancer agents exhibit the activity of both parent anticancer compounds and can overcome drug efflux effectively due to covalent binding to intracellular biomolecules, including proteins. This study demon-strates the potential applications of platinum pharmacophores as protein-targeting motifs in drug design.

## Results

### Design of platinum-based drug conjugates

We observed that clinically used anticancer platinum agents, such as cisplatin and oxaliplatin, exhibited high intracellular platinum reten-tion even in efflux-high MDR cell lines (Figs. S42, 43, 47)—a result consistent with prior observations[16,17]. We hypothesized that the intrinsic ability of these platinum compounds to circumvent drug efflux could be attributed to promiscuous covalent interactions of platinum with proteins and peptides, known chemical targets of

platinum agents[17–30], in addition to the canonical therapeutic target, DNA[31–33]. If platinum-protein adducts exceed the substrate size limit of ABC transporters[6,34], they are unlikely to be removed by efflux pumps (Fig. 1b). We thus envisioned that conjugating platinum pharmaco-phores to other drugs might mitigate efflux ability.

We chose to apply this strategy to functionalize doxorubicin, a widely used chemotherapeutic and a prototypical ABC transporter substrate, with an oxaliplatin-like pharmacophore via a synthetic route designed to avoid significant structural alterations to the backbone of the parental molecules. The synthesis started with stereoselective conversion of the *cis*−3'-amino-4'-hydroxy motif of doxorubicin to the *trans*-3',4'-diamine (Fig. 1c, see SI for synthetic details). This bidentate ligand (DoxNH₂NH₂, **1**) allows for covalent attachment of an oxaliplatin-like platinum pharmacophore, affording a drug conjugate, doxaliplatin (DoxPt, **2**), that shares key structural features of both doxorubicin and oxaliplatin. This compound differs significantly from typical platinum-containing drug conjugates, where a second bioactive component is designed to dissociate from platinum upon entering the cell[35]. For DoxPt, the platinum core and anthracycline are attached with non-leaving group NH₂ ligands[36], rendering a metal complex that likely acts as a single molecular entity rather than a simple physical mixture of parental compounds. As confirmed by HPLC studies, although DoxPt reacts with (L)-cysteine and glutathione in a pH 7.0 buffer at

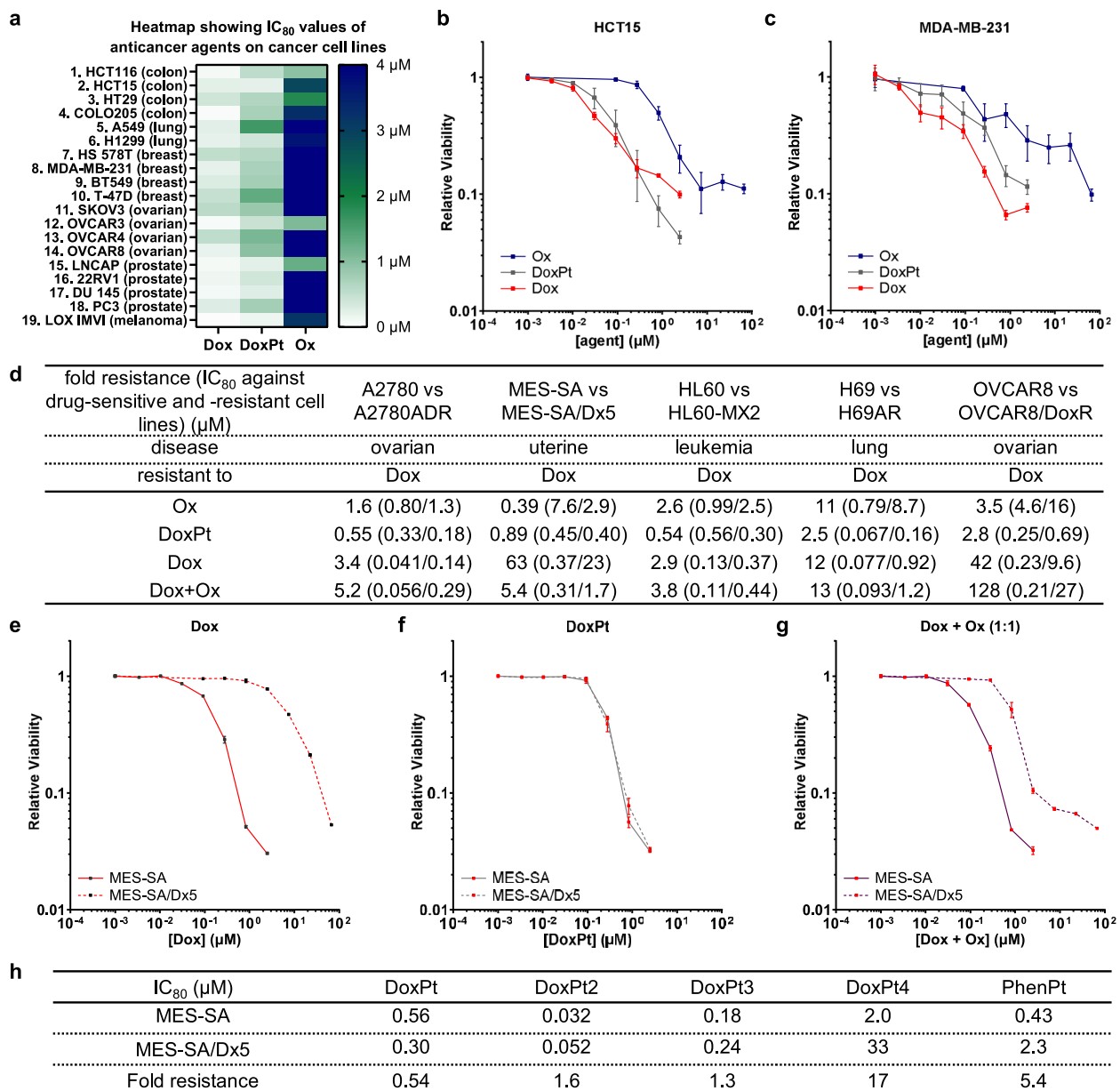

**Fig. 2 | Cytotoxicity of DoxPt against different cancer cell lines. a** Heatmap comparing cytotoxicity of oxaliplatin (Ox), doxorubicin (Dox), and doxaliplatin (DoxPt) against human cancer cell lines. $IC_{80}$ values are reported in SI (Table S1, Fig. S48-66). **b** Dose-response curves ($n = 4$ biological replicates) for Ox (blue), Dox (red), and DoxPt (gray) with HCT15 human colorectal adenocarcinoma cells. **c** Dose-response curves ($n = 4$ biological replicates) for Ox (blue), Dox (red), and DoxPt (gray) with MDA-MB-231 human breast adenocarcinoma cells. **d** Comparison of Ox, DoxPt, Dox, and a physical mixture of Dox and Ox in 1:1 molar ratio; $IC_{80}$ values for parental and drug-resistant cell lines are shown in parentheses. Fold resistance was calculated as $IC_{80\text{-resistant}} \div IC_{80\text{-parental}}$. **e** Dose-response curves ($n = 3$ biological replicates) for Dox with MES-SA human uterine sarcoma cells (solid red line) and the doxorubicin-resistant variant MES-SA/Dx5 (dashed red line). **f** Dose-response curves ($n = 3$ biological replicates) for DoxPt with MES-SA cells (solid gray line) and MES-SA/Dx5 cells (dashed gray line). **g** Dose-response curves ($n = 3$ biological replicates) for a physical mixture of Dox and Ox (1:1 molar ratio) with MES-SA cells (solid purple line) and MES-SA/Dx5 cells (dashed purple line). For all curves, the error bars indicate standard deviation. **h** Comparison of activity of different platinum-containing anticancer agents against MES-SA and MES-SA/Dx5 cells. Fold resistance was calculated as $IC_{80\text{-resistant}} \div IC_{80\text{-parental}}$. Data in panels (**b**, **c**, **e**, **f**) and (**g**) are presented as mean values ± SD. Source data are provided as a file.

37 °C to form multiple species that warrant further investigation, the release of **1** via Pt–NH$_2$ bond dissociation is not the major depletion pathway in the presence of these biomolecules at up to 1 mM concentration (Figs. S21–23, 26, 27).

**Platinum drug conjugates overcome chemoresistance**

We established cytotoxicity profiles against a range of solid tumor cell lines (Fig. 2a); DoxPt generally displayed a potency either similar to doxorubicin (Fig. 2b, HCT15) or, more commonly, intermediate between doxorubicin and oxaliplatin (Fig. 2c). We investigated the

ability of DoxPt to overcome chemoresistance, particularly ABC transporter-mediated drug efflux. An array of human cancer cell line pairs consisting of drug-sensitive lines and the corresponding ABC transporter overexpressing derivatives were treated with doxorubicin, oxaliplatin, DoxPt, or a physical mixture of the two parental drugs (Fig. 2d). We first examined the activity of DoxPt using the A2780 ovarian cancer cell line and the P-gp-overexpressing A2780ADR variant[8,37]. Although the A2780ADR line was resistant to doxorubicin treatment, minimal resistance to DoxPt was observed (Fig. S67), supporting the paradigm of using platinum to circumvent drug efflux. An

even more striking effect was observed with the MES-SA uterine sarcoma cell line and the MES-SA/Dx5 drug-resistant counterpart[38,39]. Although MES-SA/Dx5 cells exhibited more than 60-fold resistance to doxorubicin (Figs. 2e, S68), DoxPt completely abrogated drug resistance (Fig. 2f). The resistant cells, however, were insensitive to a physical mixture of equimolar doxorubicin and oxaliplatin, implicating the need for covalent incorporation of platinum (Fig. 2g). Encouragingly, DoxPt killed not only these two resistant lines but also a broad range of chemoresistant cells, revealing the general activity of this conjugate against MDR (Figs. 2h, S69, 70, 72). We observed that MRP1-promoted doxorubicin resistance in H69AR cells[40,41] was also overcome by DoxPt (Figs. 2h, S70), a result consistent with the design principle (Fig. 1b). Taken together, these findings support the hypothesis that platinum drug conjugates can evade drug clearance mediated by various ABC transporters and do so without requiring the use of an additional efflux-inhibiting agent.

To assess the effects of leaving group ligands[36] on anticancer activity, we synthesized other platinum drug conjugates (Fig. 1d) containing either the cisplatin- (**3**, DoxPt2) or carboplatin-like (**4**, DoxPt3) pharmacophore. These compounds exhibited different $IC_{80}$ values, which suggests that the leaving group ligands affect absolute potency, a phenomenon also observed with conventional platinum agents[36]. Nonetheless, both compounds are highly effective against MDR, demonstrating the role of non-leaving group ligands in overcoming efflux (Figs. S74, 75). To probe this relationship, we prepared a doxorubicin-cisplatin conjugate (DoxPt4, Fig. 1d) as a control, in which a single $Pt-NH_2$ bond tethers the two fragments. In this way, the conjugate can dissociate relatively easily, a fact confirmed by HPLC studies that show the slow release of doxorubicin from DoxPt4 in the presence of (L)-cysteine or glutathione (Figs. S31, 32). In comparison to the efflux pump substrate doxorubicin alone and the physical mixture with cisplatin (Fig. S76), DoxPt4 still partially overcomes drug efflux (Fig. 2h). In addition, phenanthriplatin (PhenPt, Fig. S77)[42], like cisplatin, carboplatin, and oxaliplatin, is also a relatively weak efflux pump substrate (Fig. 2h). These data collectively suggest an approach for augmenting conventional chemotherapeutics with platinum to overcome MDR.

## DoxPt is a poor efflux pump substrate

We sought to determine whether DoxPt kills P-gp-overexpressing cells independently of P-gp activity, and the importance of platinum-protein covalent interactions in the process. We began by treating MES-SA and MES-SA/Dx5 cells with doxorubicin in the presence of verapamil, a commonly used P-gp inhibitor[43]. As expected, we observed a complete reversal of doxorubicin resistance in MES-SA/Dx5 cells (Figs. 3a, S78). In contrast, verapamil did not enhance the activity of DoxPt against either MES-SA or MES-SA/Dx5 cells, indicating that DoxPt is not a P-gp substrate (Figs. 3b, S80). The ability of DoxPt to circumvent ABC transporter activity is evident from measurements of intracellular platinum content using atomic absorption spectroscopy (AAS). Platinum clearance of DoxPt was independent of baseline MDR efflux activity levels, as adduced from the similar platinum retention in MES-SA and MES-SA/Dx5 cells (Figs. 3c, S42). Platinum accumulation was also largely unaffected by verapamil treatment (Figs. 3d, S43), a result consistent with in vitro toxicity assays (Fig. 3b). As anticipated, the behavior of DoxPt mimicked that of oxaliplatin (Figs. 3c, d, S42, 43), with similar and long-lasting platinum retention regardless of efflux activity levels. These data substantiate platinum as a modulator of cellular retention.

We examined the accumulation and distribution of doxorubicin and DoxPt, which are intrinsically fluorescent, in live cell imaging experiments. Doxorubicin fluorescence was observed in the nuclei of MES-SA cells but was absent from most of the nuclei of MES-SA/Dx5 cells, a phenomenon that was reversible with verapamil treatment (Figs. 3e, f, S87, 88, 92). In contrast, both DoxPt-treated MES-SA and MES-SA/Dx5 cells exhibited similar fluorescence, the intensity of which

was not significantly affected by verapamil (Figs. 3g, S89, 90, 92). These results provide corroborating evidence that DoxPt is not efficiently removed by P-gp. Additionally, DoxPt and doxorubicin exhibited different subcellular distribution patterns. Although doxorubicin predominantly accumulated in the nucleus, DoxPt was mainly found in the cytoplasm (Figs. 3g, S93–97). Subsequent AAS-based quantification showed similar results: MES-SA and MES-SA/Dx5 cells exposed to oxaliplatin or DoxPt had nearly identical platinum subcellular distribution (Fig. 3h). For both compounds, more than 60% of the platinum accumulated in the cytoplasm and about 10% in the nucleus (Fig. S44). Protein and nucleic acid fractionations of cells treated with oxaliplatin or DoxPt were quantified by AAS. Although each sample contained platinum, the majority was found in the protein fractions, with DoxPt exhibiting a higher protein preference than oxaliplatin (Figs. 3i, S45).

We examined the reactivity of DoxPt with proteins by gel electrophoresis using bovine serum albumin (BSA) as a model substrate. Compared to doxorubicin and $DoxNH_2NH_2$, DoxPt showed the highest degree of binding to BSA (Figs. S38, 39). With MES-SA cell lysate as a model system, DoxPt associated with a large number of proteins (Figs. 3j, S40, 41). In contrast, Dox and $DoxNH_2NH_2$ did not exhibit significant affinity towards proteins. Together, these experiments demonstrate how platinum incorporation can enhance drug conjugate-protein interactions to evade efflux activity and thus overcome chemoresistance.

To understand cytotoxic mechanisms, we investigated the activity of DoxPt using the NCI60 screening panel[44]. With the pattern comparison tool in CellMiner[45], we found strong correlations for DoxPt and classical $N^7$-alkylating agents, including carboplatin, cisplatin, and oxaliplatin, as well as Top2 inhibitors, such as etoposide and anthracyclines (Table S5). The activity of DoxPt also moderately correlated with irinotecan, topotecan, and other Top1 inhibitors. These findings suggest that, in addition to retaining parental drug activity, DoxPt may also target Top1, which would account for DoxPt resistance in CEM/C2 cells with a Top1 mutation (Fig. S71)[46,47]. We also conducted an established RNAi-based competition assay in Eμ-Myc lymphoma cells[48–50]. This functional genetic method employs GFP-tagged short hairpin RNAs (shRNAs) to target one of eight genes encoding proteins involved in cell-death signaling pathways. Upon drug treatment, measurements of the enrichment or depletion of cells expressing each of the eight shRNAs compared to non-expressing controls constitute drug signatures, which classify cytotoxic agents by mechanism of action. Using this platform, we compared the signature of DoxPt to that of Dox, Ox, and a physical mixture of Dox and Ox at equipotent doses. The resistance and sensitivity patterns of DoxPt are most similar to those of the Dox-Ox physical mixture, which involves several DNA-damaging signatures, such as strong *p53*- and *Chk2*-dependence (Figs. 3k, S98). These experiments provide additional evidence that DoxPt retains the activity of both parental compounds. Despite the similarities, in a principal component analysis, DoxPt clusters in a class separate from existing single drug-based categories in the reference set (Fig. S99). We validated the biomolecular targets of DoxPt using a Top2α decatenation assay (Table S6 and Figs. S101, 102) and a Top1 intercalation assay (Figs. S106–108). Like doxorubicin, DoxPt inhibited both enzymes, but to a lesser extent.

## DoxPt delays the acquisition of resistance

To assess the activity of DoxPt, particularly the ability to overcome drug resistance, we chose cancer organoids as more physiologically relevant models. We developed colorectal cancer organoids that have mutations commonly found in colon cancer, generating $APC^{-/-}$; $KRAS^{G12D}$; $p53^{-/-}$ (AKP) organoids and $APC^{-/-}$; $KRAS^{G12D}$; $p53^{-/-}$; $SMAD4^{-/-}$ (AKPS) organoids. These organoids were also engineered to express either TdTomato (TdT) or ZsGreen (ZsG) to facilitate fluorescence microscopy-based in vitro co-culture studies. In this system, DoxPt

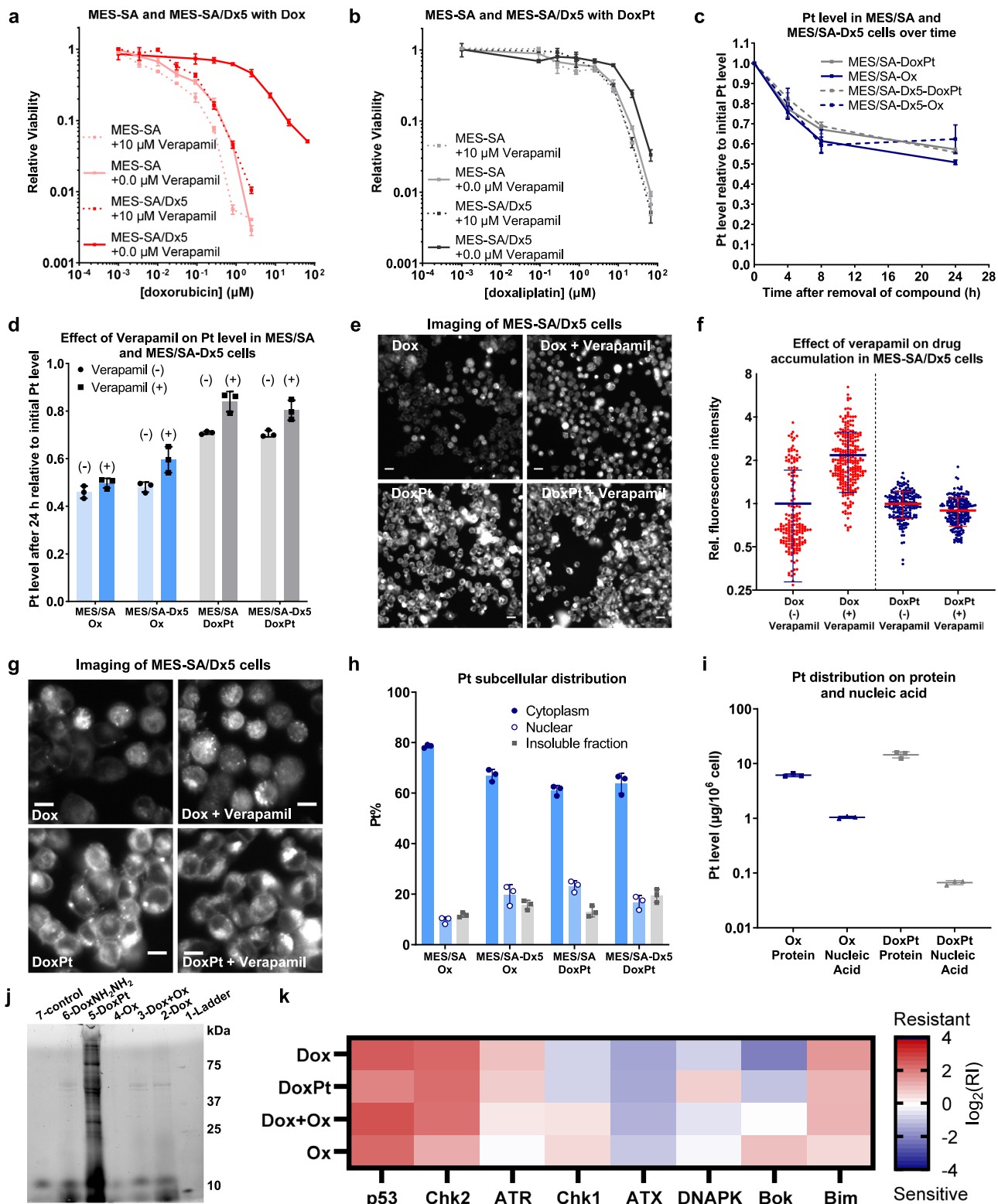

exhibited an intermediate potency between doxorubicin and oxaliplatin (Fig. S83). We also assessed the rate of resistance acquisition of different anticancer agents. Here, we exposed AKP and AKPS organoids to escalating dose responses of doxorubicin, oxaliplatin, or DoxPt and passaged them weekly at the highest drug concentration that allowed for outgrowth. After five months of doxorubicin selection, the organoids acquired more than 200-fold doxorubicin resistance (Fig. 4a), which was substantially reversed by verapamil, implicating a P-gp-driven resistance mechanism (Fig. 4b, c). In stark contrast, even

after five months of selection, DoxPt induced only minimal drug resistance (Figs. 4a, S84). As predicted, DoxPt also effectively killed the doxorubicin-resistant AKP/DoxR organoids (Fig. 4b). As seen with OVCAR cell lines (Figs. S72, 73), oxaliplatin-resistant AKP/OxR organoids did not confer resistance to DoxPt (Fig. 4d). Finally, mildly DoxPt-resistant AKP/DoxPtR organoids exhibited little cross-resistance to doxorubicin or oxaliplatin (Fig. 4e), indicating that DoxPt-induced resistance is likely mechanistically different from the parental compounds.

**Fig. 3 | Investigation of the modes of action of DoxPt. a** Dose-response curves showing the effect of verapamil on doxorubicin-treated MES-SA and MES-SA/Dx5 cells ($n = 3$ biological replicates). **b** Dose-response curves showing the effect of verapamil on DoxPt-treated MES-SA and MES-SA/Dx5 cells ($n = 3$ biological replicates). **c** Pt clearance from MES-SA and MES-SA/Dx5 cells treated with oxaliplatin (10.0 μM) or DoxPt (1.25 μM) for 12 h ($n = 3$ biological replicates). The horizontal axis indicates time after the cells were switched to drug-free media. **d** Effect of verapamil on Pt clearance from MES-SA and MES-SA/Dx5 cells treated with oxaliplatin (10.0 μM) or DoxPt (1.25 μM) for 12 h. The Pt levels were determined at 0 h and 24 h after the cells were switched to drug-free media ($n = 3$ biological replicates). **e** Representative fluorescence images showing the effect of verapamil on drug accumulation in MES-SA/Dx5 cells treated with Dox (2.5 μM) or DoxPt (2.5 μM) for 18 h. Scale bar = 20 μm. **f** Quantification of fluorescence microscopy images. For each agent, fluorescence is normalized to the average absolute intensity of cells treated in the absence of verapamil. The results are consistent across three independent experiments. One experiment was chosen for quantification ($n = 183$ cells for Dox (−) verapamil, $n = 262$ cells for Dox (+) verapamil, $n = 192$ cells for DoxPt (−) verapamil, $n = 205$ cells for Dox (+) verapamil). **g** Representative fluorescence images showing the subcellular distribution of Dox (2.5 μM) or DoxPt (2.5 μM) in MES-SA/Dx5 cells after 18 h treatment. Scale bar = 10 μm. **h** Subcellular Pt distribution in MES-SA and MES-SA/Dx5 cells treated with oxaliplatin (10.0 μM) or DoxPt (1.25 μM) for 14 h ($n = 3$ biological replicates). **i** Pt levels in protein and nucleic acid fractions from MES-SA cells treated with oxaliplatin (25.0 μM) or DoxPt (5.0 μM) for 18 h ($n = 3$ biological replicates). **j** Fluorescence gel image of MES-SA cell lysate treated with the indicated compounds. The qualitative gel is representative of three independent experiments. **k** A heat map showing the signature of Dox, DoxPt, an equipotent Dox+Ox physical mixture, and Ox. Data in panels (**a**–**d**, **f**, **h**) and (**i**) are presented as mean values ± SD. Source data are provided as a file.

To assess the activity of different anticancer agents under identical conditions, we conducted co-culture experiments with red fluorescent doxorubicin-resistant AKP/DoxR-TdT organoids and green fluorescent doxorubicin-sensitive AKP-ZsG organoids. As shown in Fig. 4f, although neither doxorubicin nor a physical mixture of doxorubicin and oxaliplatin effectively killed AKP/DoxR-TdT organoids, DoxPt exhibited a high activity against both drug-sensitive and drug-resistant organoids. These results collectively demonstrate the effectiveness of DoxPt in a variety of drug-resistance contexts, which cannot be achieved by a physical mixture of the parental compounds. We also examined the in vivo efficacy of DoxPt in a proof-of-principle experiment with a mouse model of colon cancer peritoneal carcinomatosis using the hyper-aggressive AKPS-TdT organoids. Weekly administration of DoxPt significantly extended the median survival of mice from 22.5 days to 34 days (Fig. 4g). In a separate experiment, we verified that a three-week course of DoxPt treatment also effectively reduced the tumor burden in this model (Fig. 4h).

In conclusion, we demonstrated an effective strategy to employ protein-reactive platinum functionalities to generate drug conjugates, which augment traditional cytotoxic chemotherapeutic agents. We mechanistically delineated how platinum conjugation can subvert undesired efflux to overcome MDR. Furthermore, we found that the conjugates retain several desirable properties of platinum agents and anthracyclines. Although DNA has long been pursued as the primary therapeutic target for many conventional platinum agents, we propose that underexplored covalent platinum-protein interactions can be employed to develop therapeutics for resistant cancers. Given the broad clinical use of platinum drugs and the widely tunable reactivity of platinum pharmacophores towards proteins, we anticipate that this approach may be applicable to next-generation metallodrugs.

## Methods

The research described complies with all relevant ethical regulations. Mice were under the husbandry care of the Department of Comparative Medicine in the Koch Institute for Integrative Cancer Research. C57BL/6 mice from the Jackson Laboratory were used for in vivo studies. Only female mice at the ages of three to five months were used to minimize variability of animal body mass. A total number of 39 animals were used. Mice were housed under a 12-h light–dark cycle at $21 \pm 1\,°C$ with humidity of $50 \pm 10\%$. Animals were co-housed with littermates with ad libitum access to water and food. Mice were never allowed to bear a tumor burden exceeding the maximal tumor size of 1 cm in diameter, as approved by MIT's Committee on Animal Care. All procedures were conducted in accordance with the American Association for Accreditation of Laboratory Animal Care and approved by MIT's Committee on Animal Care.

### Compounds used for cytotoxicity studies

Pharmaceutical-grade cisplatin (catalog # PHR1624), oxaliplatin (catalog # PHR1528), carboplatin (catalog # PHR3417), and doxorubicin hydrochloride (catalog # D1515) were purchased from Sigma-Aldrich. Doxorubicin stock solutions were prepared by dissolving doxorubicin hydrochloride in $H_2O$ and stored at $-20\,°C$. The stock solution of doxorubicin was used and discarded after each experiment. Oxaliplatin and carboplatin stock solutions were prepared by dissolving the corresponding platinum agent in Milli-Q water ($18\,M\Omega\,cm^{-1}$) at rt and stored at $-20\,°C$. Cisplatin stock solutions were prepared by dissolving cisplatin in 0.9 wt% sodium chloride solution at rt and stored at $-20\,°C$. DoxPt and DoxPt3 stock solutions were prepared by dissolving the corresponding compound in a glycerol/water mixture (1:1 v/v). The suspension was heated at $60\,°C$ for about 5 min to facilitate solubilization. The stock solutions were stored at $-20\,°C$ and heated at $60\,°C$ for 1–2 min to dissolve the precipitated compound before use. DoxPt2 solutions were prepared in glycerol. The suspension was heated at $60\,°C$ for about 5 min to facilitate solubilization. This solution was then diluted with an equal volume of 0.9 wt % sodium chloride solution. This solution was stored at $-20\,°C$ and heated at $60\,°C$ for 1–2 min to dissolve the precipitated compound before use. All platinum agent stock solutions were filtered through a sterile 0.2 μm regenerated cellulose syringe filter after preparation. The concentration of the filtrate was determined by atomic absorption spectrometry. Verapamil (Sigma-Aldrich catalog # V4629) was prepared as a 10 mM stock solution in 100% EtOH.

### Cell lines used in this study

Unless stated otherwise, cell lines were obtained from the Robert A. Swanson (1969) Biotechnology Center at the Massachusetts Institute of Technology. Cell lines, including A2780ADR (catalog # 93112520), MES-SA (catalog # 95051030), and MES-SA/Dx5 (catalog # 95051031), were purchased from Sigma-Aldrich. Drug-sensitive and -resistant cell line pairs, including HL-60 (Catalog # CRL-240) and HL-60/MX2 (Catalog # CRL-2257), H69 (Catalog # HTB-119) and H69AR (Catalog # CRL-11351), and CEM (Catalog # CCL-119) and CEM/C2 (Catalog # CRL-2264), were purchased from ATCC. The NCI-60 Human tumor cell lines screen was conducted by the Development Therapeutics Program at the National Cancer Institute using 60 human cancer cell lines. Cell lines from ATCC and Sigma-Aldrich were authenticated by the manufacturers using short tandem repeat (STR) profiling. These data can be found on the manufacturers' websites. Cell lines from the Robert A. Swanson (1969) Biotechnology Center were authenticated by ATCC with STR profiling. NCI-60 Human tumor cell lines were extensively authenticated by NCI with various methods, including STR profiling. The Eμ-Myc cell line was generated in-house and was not authenticated independently in this study.

The oxaliplatin-resistant OVCAR8 cell line (OVCAR8/OxR) was generated by treating the OVCAR8 human ovarian cancer cell line repeatedly with increasing concentrations of oxaliplatin until at least a 10-fold increase in $IC_{50}$ was observed. The doxorubicin-resistant OVCAR8 cell line (OVCAR8/DoxR) was generated by treating the

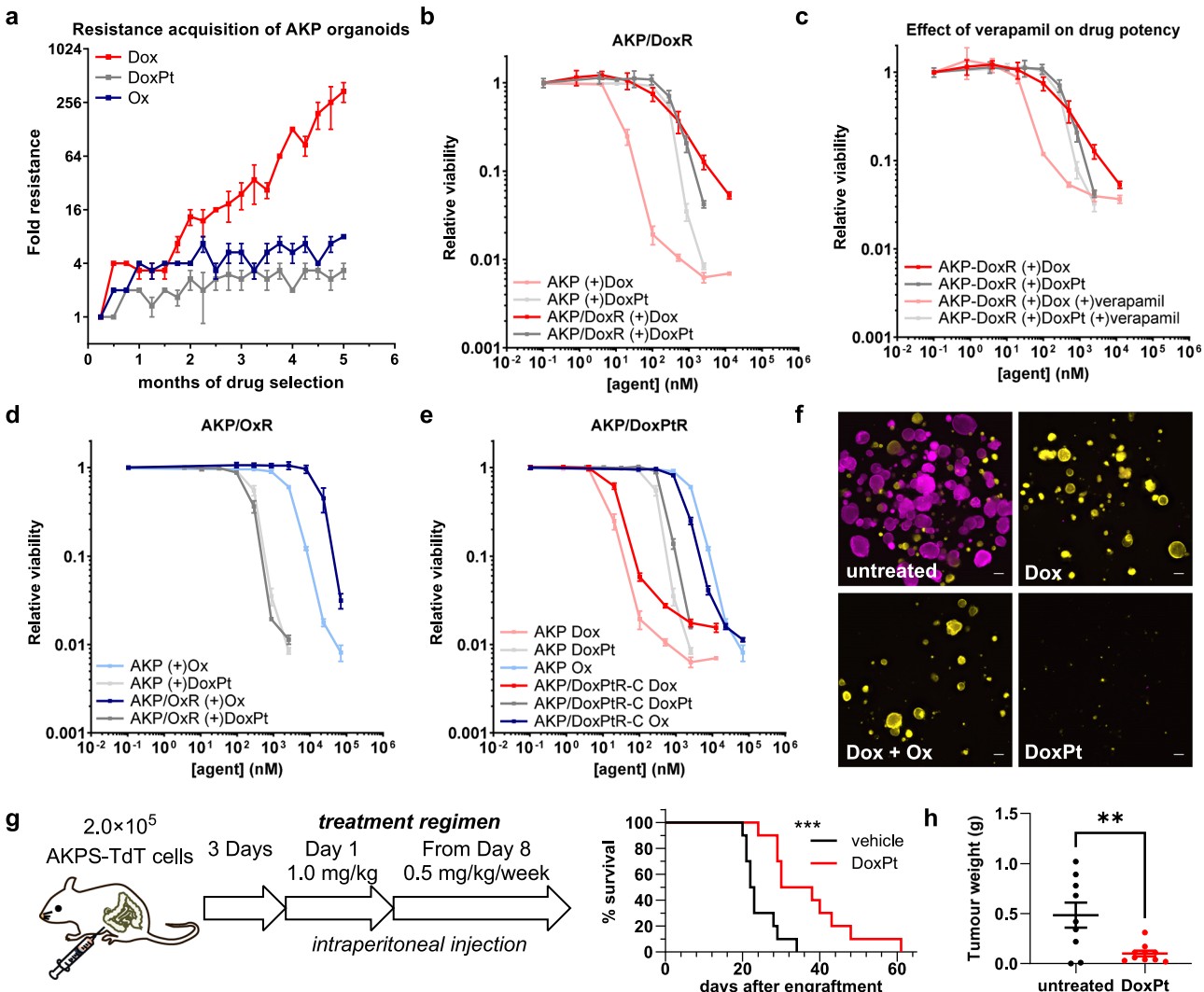

**Fig. 4 | Anticancer activity of DoxPt with organoid-based in vitro and in vivo models. a** Resistance acquisition by *APC⁻/⁻; KRAS^{G12D}; p53⁻/⁻* (AKP) murine colorectal cancer organoids after five months of drug selection (Dox, red; Ox, blue; DoxPt, gray, n = 3 biological replicates for each drug selection). **b** Dose-response curves showing the effect of Dox (red) and DoxPt (gray) on AKP cancer organoids and doxorubicin-resistant AKP/DoxR cancer organoids (n = 3 biological replicates). **c** Effect of verapamil on AKP and AKP/DoxR cancer organoids treated with doxorubicin or DoxPt (n = 3 biological replicates). **d** Effect of Ox (blue) and DoxPt (gray) on AKP cancer organoids and oxaliplatin-resistant AKP/OxR cancer organoids (n = 3 biological replicates). **e** Dose-response curves showing the effect of Dox (red), Ox (blue), and DoxPt (gray) on AKP and AKP/DoxPtR cancer organoids (n = 3 biological replicates). **f** Representative fluorescence microscopic images showing the co-cultured AKP-TdT (pink) and AKP/DoxR-ZsG (yellow) cancer organoids treated with

Dox (1.25 μM), a physical mixture of Dox+Ox (both at 1.25 μM), or DoxPt (1.25 μM) for 7 d. Scale bar = 200 μm. **g** DoxPt treatment of a hyper-aggressive AKPS organoid-based colon cancer peritoneal carcinomatosis mouse model. Median survival times of DoxPt-treated mice and the control group were 34 days and 22.5 days, respectively. n = 10 mice per treatment group. ***P = 0.0005 by two-sided Log-rank test. **h** Tumor burden reduction from DoxPt treatment in the AKPS organoid-based colon cancer peritoneal carcinomatosis mouse model. Treatment was started three days after engraftment with the following regimen: 2 mg/kg on Day 3, 1 mg/kg on Day 10 and Day 17, respectively. Mice were euthanised on Day 23, and tumors were dissected and weighed. n = 10 mice for DoxPt-treated group. n = 9 mice for the control group. **P = 0.0059 by unpaired two-tailed Student's t test. Data in panels a, b, c, d, e, and h are presented as mean values ± SD. Source data are provided as a file.

OVCAR8 human ovarian cancer cell line repeatedly with increasing concentrations of doxorubicin until at least a 50-fold increase in IC$_{50}$ was observed.

Cells were grown in media (10 mL) in 10 cm tissue culture plates. The cells were trypsinized and seeded into 96-well plates with $5 \times 10^3$ cells per well. Cells were treated with cytotoxic drug-containing media (200 μL) at the time of plating. For experiments with P-gp inhibition, verapamil was applied at the indicated concentration at the time of cell seeding along with the cytotoxic agent. The cells were then incubated for 72 h following the drug treatment. Cell growth was measured by a resazurin assay. In brief, at the end of 72 h drug treatment, resazurin was added to the cells at a final concentration of 50 μg/mL. The baseline fluorescence was determined immediately. The cells were

then incubated at 37 °C for 2 h. The fluorescence of the resorufin product was measured at 560/590 nm and baseline corrected. The cell growth at each drug concentration was normalized to the untreated control.

### Colorectal cancer organoid generation and related cytotoxicity studies

Colonic crypts were isolated and colon organoids were generated from the following mouse genotypes on a C57Bl/6 background: 1. Rosa-LSL-TdTomato (hereafter LSL-TdT), 2. P53^{fl/fl}; LSL-TdT, and 3. LSL-KRAS^{G12D}; p53^{fl/fl}; LSL-TdT. Subsequently, the pSECC-APC plasmid (carrying Cre, Cas9 and sgAPC)[7] was transfected using Lipofectamine™ 2000 (ThermoFisher catalog # 11668019) into each of the above three genotypes

of colon organoids. This procedure was generated in a single step 1. $APC^{-/-}$; $TdT^+$ (A-TdT), 2. $APC^{-/-}$; $p53^{-/-}$; $TdT^+$ (AP-TdT), and 3. $APC^{-/-}$; $KRAS^{G12D/+}$; $p53^{-/-}$; $TdT^+$ (AKP-TdT) colorectal cancer organoids, which were selected with Wnt withdrawal. The $APC^{-/-}$; $KRAS^{G12D/+}$; $p53^{-/-}$; $SMAD4^{-/-}$; $TdT^+$ (AKPS-TdT) organoids were generated from the AKP-TdT organoids with CRISPR deletion of SMAD4 and subsequent TGF-β1 selection (PeproTech catalog #: 100-21). The $APC^{-/-}$; $KRAS^{G12D/+}$; $p53^{-/-}$; $ZsGreen^+$ (AKP-ZsG) colorectal cancer organoids were generated in a similar manner using pSECC-APC starting with LSL-Kras$^{G12D}$; p53$^{fl/fl}$; LSL-ZsG organoids.

To determine IC$_{50}$ values for cytotoxic agents, trypsinized organoids were seeded in 10 μL drops containing 67 v/v% Matrigel® in 48-well plates and incubated at 37 °C for 15 min. Drug-containing media (300 μL) were then added. For experiments with P-gp inhibition, verapamil was applied at the indicated concentration at the time of the cytotoxic agent treatment. The organoids were incubated for 72–96 h following the treatment. Organoid growth was measured by a resazurin assay. In brief, at the end of drug treatment, resazurin was added to the organoid-containing wells at a final concentration of 50 μg/mL. The baseline fluorescence was determined immediately. The organoids were then incubated at 37 °C for 2 h. The fluorescence of the resorufin product was measured at 560/590 nm and baseline corrected. The growth at each drug concentration was normalized to the untreated control.

### Resistant colorectal cancer organoid generation and related cytotoxicity studies

Drug-resistant AKP-TdT organoid lines were generated via iterative rounds of drug selection with escalating concentrations. Briefly, organoids were seeded in seven Matrigel® droplets (67 v/v%, 10 μL) per well on a 12-well plate. The organoids were treated with cytotoxic agents at three different concentrations (1×, 2×, and 4× of the IC$_{80}$ values determined from non-resistant organoids). After one week of drug treatment, organoids from the well with the highest drug concentration permitting outgrowth were collected, trypsinized, and seeded into four wells on a 12-well plate with seven Matrigel® droplets per well. Media containing the cytotoxic agent were added to wells at three concentrations (0.5×, 1×, and 2× of the previous highest outgrowth concentration). The fourth well was left drug-free as a backup. This procedure was repeated for 20 passages. The highest concentration permitting outgrowth was recorded at each passage. This experiment was performed in three independent replicate lines for each drug. Drug-resistant AKPS-TdT organoid lines were generated in a similar manner.

### In vivo studies

For in vivo mouse experiments, metastatic AKPS-TdT colon cancer organoid lines were used. Trypsinized organoids (2.0 × 10$^5$ cells) were seeded in the peritoneum of C57BL/6 mice (the Jackson Laboratory). For in vivo dosing, DoxPt was dissolved in glycerol and water (50:50 v/v). DoxPt treatment was initiated three days after tumor engraftment and dosed weekly thereafter via intraperitoneal injection. For the survival study, two groups of mice, each with ten animals, were used. Doxaliplatin was initially dosed at 1 mg/kg on Day 3 after tumor engraftment, and the mice were subsequently dosed at 0.5 mg/kg weekly. For the tumor burden reduction study, ten mice were used for the DoxPt-treated group, and nine mice were used for the control group. Doxaliplatin was initially dosed at 2 mg/kg on Day 3 after tumor engraftment, and the mice were subsequently dosed at 1 mg/kg on Day 10 and Day 17. The mice were euthanized on Day 23, and tumors were dissected and weighed. All experiments involving mice were carried out using young adult female mice (3 to 5 months old). Only female mice were used to minimize variability in animal body mass. Mice were housed under a 12-h light–dark cycle at 21 ± 1 °C with humidity of 50 ± 10%. Animals were co-housed with littermates with ad libitum access to water and food. Mice were never allowed to bear a tumor burden exceeding the maximal tumor size of 1 cm in diameter, as approved by MIT's Committee on Animal Care.

### Characterization of DoxPt with RNAi-based competition assay

A panel of eight previously validated shRNAs was delivered to Eμ-Myc Cdkn2a$^{Arf-/-}$ lymphoma cells using the pMSCV-LTR-miR30-SV40-GFP (MLS) retroviral vector. The lymphoma cells were grown in B-cell media (BCM), which was composed of Dulbecco's Modification of Eagle's Medium (DMEM, Corning catalog # 45000-306) and Iscove's Modification of DMEM (Corning catalog # 45000-366) supplemented with 10% fetal bovine serum (Gibco catalog # A2720803), 1% penicillin/streptomycin (Corning catalog # 45000-652), and 0.1% 2-mercaptoethanol (Gibco catalog # 21985023). The cells were infected with each of the GFP-tagged shRNA constructs at 25–30% infection rate as previously described[48]. Cells expressing each shRNA were seeded into 24-well plates with 250 μL BCM (125,000 cells/well for treatment wells and 65,000 cells/well for control wells). Drug-containing media (250 μL) was then added to the cells. For control wells, 300 μL of media was removed and replaced with 300 μL of fresh BCM media after 24 h. An additional 500 μL of fresh BCM media was then added to all wells. After 48 h of drug treatment, the GFP percentage of treatment and control wells was determined by flow cytometry using a BD FACSCelesta™ cell analyzer, with live cells quantified based on DAPI exclusion. The experiment was performed independently multiple times, each time with three technical replicates. For each drug, the resistance index (RI) was calculated for cells expressing each shRNA as previously defined[8]. The RI values for generating the signature of each treatment in the heat map (Fig. 3k) are listed in Table S3. Subsequently, each signature was compared to an established reference set of drugs (Table S4) using the modified K-nearest neighbors (K-NN) algorithm with reported parameters and code. This algorithm classifies the drug of interest into the nearest category in the reference set using Euclidean K-NN analysis. The linkage ratio (LR) is then calculated by dividing the pairwise distances of the new drug-containing category by the category without the new drug. Next, the LR values for all out-of-category drugs are calculated assuming that these drugs are members of that specific category, which provides a background distribution of negative-control LRs. Lastly, the $p$-value was obtained by comparing the LR of the drug of interest to the distribution of negative-control LRs. When the $p$-value is greater than 0.05, the drug of interest is considered to belong to a new drug class with a mechanism of action not represented in the reference set[45]. Additionally, principle component analysis (PCA) using the "pca" function in MATLAB was also performed to visualize the drug classification.

### Reporting summary

Further information on research design is available in the Nature Portfolio Reporting Summary linked to this article.

## Data availability

All data are available from the corresponding authors upon request. Experimental details, procedures, NMR spectra, mass spectrometry data, HPLC analysis, metal content determination, cytotoxicity studies, gel electrophoresis studies, NCI60 analysis, biochemistry and RNAi assays, and microscopy images are provided in the Supplementary Information. Source data are provided with this paper.

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

## Acknowledgments

This work was supported by National Institute of Health grant NIH/NCI CA034992 to O.H.Y. and S.J.L. and the MIT Stem Cell Initiative, NIH/NCI CA250554, and the Emerson Collective Cancer Research Fund to O.H.Y. This work was also supported by the Koch Institute Frontier Research Fund to O.H.Y. and M.T.H. We thank the Swanson Biotechnology Center for equipment, resources, and technical support. F.W. thanks the Rhode Island Foundation for a Medical Research Grant and support of the Rhode Island Life Science Hub. The Rhode Island Institutional Development Award (IDeA) Network of Biomedical Research Excellence from the National Institute of General Medical Sciences of the National Institutes of Health under Grant Number P20GM103430 is acknowledged for support through the Centralized Research Core Facility and the EDC award to F.W. N.L.P. acknowledges the University of Rhode Island for internship funding. We thank William C. Reinhold at the Genomics and Pharmacology Facility of the Developmental Therapeutics Branch, NCI, NIH, for assistance with the NCI60 analysis.

## Author contributions

F.W., J.B., S.J.L. and Ö.H.Y. conceived the work, F.W., J.B., G.E., J.M.G., M.T.H. and Ö.H.Y. designed the experiments, F.W., J.B., G.E., O.L., N.L.P., D.S.H, S.I., B.P., D.Z., J.M.M., C.S.H., A.A.G., W.K.M. and J.M.G. conducted the experiments, F.W., J.B., J.M.G., M.T.H., S.J.L. and Ö.H.Y. supervised the research, and F.W., J.B., G.E., O.L., N.L.P., J.M.G., S.J.L. and Ö.H.Y. wrote the manuscript.

## Competing interests

F.W., J.B., S.J.L. and Ö.H.Y. declare the following competing financial interest(s): Patent applications have been filed for this work. The remaining authors declare no competing interests.
