## [Transparent Peer Review file · Nature Communications]

Leveraging platinum-protein interactions to overcome chemoresistance

Corresponding Author: Professor Omer Yilmaz

Version 0:

Reviewer comments:

Reviewer #1

(Remarks to the Author)

Wang et al. have developed an innovative chemical strategy to overcome resistance to doxorubicin, which commonly arises from the overexpression of efflux pumps that reduce intracellular concentrations of doxorubicin and other anthracyclines. Briefly, the authors conjugated doxorubicin and doxorubicin-based analogues with platinum-based chemotherapeutic agents (e.g. oxaliplatin, cisplatin). This design leverages the ability of platinum-based drugs to form adducts with intracellular proteins, thereby reducing recognition and export of the drug conjugates by efflux pumps. Notably, these drug conjugates retain the mechanisms of action of both anthracycline and platinum-based drugs, while representing a novel therapeutic modality with the potential to overcome resistance to the parent compounds. Overall, the authors have sufficiently addressed the concerns of the previous reviewer. Thus, I strongly recommend the revised manuscript for publication.

Comment:

The manuscript could be further improved by including negative control samples (i.e. cells untreated with doxorubicin or DoxPt) in the imaging studies (Fig. 3g,h and Fig S80–S88). Including these controls would help the reader distinguish between anthracycline-specific fluorescence and background fluorescence that stems from cellular autofluorescence.

The authors have sufficiently addressed the concerns raised by Reviewer 1. I recommend the revised manuscript for publication.

Reviewer #2

(Remarks to the Author)

The authors have carefully addressed most of the comments from reviewer 3 in their rebuttal. However, some issues remain that need to be addressed before publication.

-The main problem concerns stability of DoxPt in cell culture medium and particularly L-cysteine. The authors report that the medium contains 100 μ M L-cysteine. They then test stability of the compound in presence of 200 μ M or 1 mM L-cysteine, resulting in quick degradation of the compound (almost no DoxPt left at 4h). While the authors claims that release of DOXNH₂NH₂ is only a trace amount, I am not sure this is the case since when it appears on the HPLC the other peaks have almost disappeared (so if we were to integrate the peaks the % may be higher than immediately apparent). In any case, whether DOXNH₂NH₂ is significantly released or not, does not change the fact that incubation with 200 μ M cysteine results in degradation of the compound. Is this also the case at the 100 μ M concentration found in medium?

If that is the case, what is the actual species that cause the observed cytotoxicity and the overcoming of resistance? The authors should try to find out what the newly formed species are as they may be responsible for the interesting properties described in the paper.

Other minor issues with the current response are described below.

-The authors write "RNAi signature assay showed that the modes of action of DoxPt resemble those of both doxorubicin and oxaliplatin" and in the main text they mention that they "compared the signature of DoxPt to that of Dox, Ox, and a physical mixture of Dox and Ox at equipotent doses." However, only the DoxPt and physical mixture data are reported in Figure S90. The Ox and Dox data should also be reported for completeness. Also, some statistically significant differences can be observed for some shRNA and require a comment on why this is (or isn't) important in determining the overall mechanism of action of the DoxPt agent.

-When reviewer 3 asks about differences in behaviour between the "released" DoXNH₂NH₂ and Dox in binding to major groove of DNA the authors responds that DONH₂NH₂ is, effectively, not released. However, this does not really answer the question about differences in behaviour compared to Dox. In fact, one could expect that, if DoxPt stayed intact, the binding to the minor groove of DNA would be even more affected, given that there would be not just a sugar but a full Pt complex attached to it. The authors should comment on this.

Version 1:

Reviewer comments:

Reviewer #2

(Remarks to the Author)

The authors have thoroughly addressed all of the reviewers' comments and repeated experiments that led to concern regarding discrepancy in the data. I particularly appreciated the inclusion of similar stability experiments with oxaliplatin to compare the stability of the two compounds in presence of cysteine. I commend the authors dedication to improving the manuscript and ensuring the robustness of its results and recommend the paper for publication.

Responses to reviewer comments

Reviewer #1 (Remarks to the Author):

Wang et al. have developed an innovative chemical strategy to overcome resistance to doxorubicin, which commonly arises from the overexpression of efflux pumps that reduce intracellular concentrations of doxorubicin and other anthracyclines. Briefly, the authors conjugated doxorubicin and doxorubicin-based analogues with platinum-based chemotherapeutic agents (e.g. oxaliplatin, cisplatin). This design leverages the ability of platinum-based drugs to form adducts with intracellular proteins, thereby reducing recognition and export of the drug conjugates by efflux pumps. Notably, these drug conjugates retain the mechanisms of action of both anthracycline and platinum-based drugs, while representing a novel therapeutic modality with the potential to overcome resistance to the parent compounds. Overall, the authors have sufficiently addressed the concerns of the previous reviewer. Thus, I strongly recommend the revised manuscript for publication.

Comment:

The manuscript could be further improved by including negative control samples (i.e. cells untreated with doxorubicin or DoxPt) in the imaging studies (Fig. 3g,h and Fig S80–S88). Including these controls would help the reader distinguish between anthracycline-specific fluorescence and background fluorescence that stems from cellular autofluorescence.

=> The images of untreated cells as negative controls have been included in the Supplementary Information (Fig. S91). These images show minimal background fluorescence under the experimental conditions.

The authors have sufficiently addressed the concerns raised by Reviewer 1. I recommend the revised manuscript for publication.

Reviewer #2 (Remarks to the Author):

The authors have carefully addressed most of the comments from reviewer 3 in their rebuttal. However, some issues remain that need to be addressed before publication.

-The main problem concerns stability of DoxPt in cell culture medium and particularly L-cysteine. The authors report that the medium contains 100 μM L-cysteine. They then test stability of the compound in presence of 200 μM or 1 mM L-cysteine, resulting in quick degradation of the compound (almost no DoxPt left at 4h). While the authors claims that release of DOXNH₂NH₂ is only a trace amount, I am not sure this is the case since when it appears on the HPLC the other peaks have almost disappeared (so if we were to integrate the peaks the % may be higher than immediately apparent). In any case, whether DOXNH₂NH₂ is significantly released or not, does not change the fact that incubation with 200 μM cysteine results in degradation of the compound. Is this also the case at the 100 μM concentration found in medium?

=> To address this question, we conducted additional experiments by monitoring the progress of the reaction of DoxPt with 100 μM L-cysteine (Fig. S14 and S21). We also repeated the experiments using 200 μM L-cysteine to address the discrepancy previously observed (Fig. S15 and S22). As indicated by HPLC, about 17% DoxPt was left after 12 h in pH 7.0 buffer containing 200 μM L-cysteine at 37 °C, as either the parent compound or in the form of DoxPt₂ due to oxalate-chloride exchange. In contrast, 37% remained this way in the reaction with 100 μM L-cysteine under the same conditions. Additionally, control studies with oxaliplatin and L-cysteine at 100 μM and 200 μM showed analogous results (Fig. S18-19). These investigations reveal similar L-cysteine reactivity of the platinum moiety in DoxPt and oxaliplatin (Fig. S20 and S29).

To quantify the DoxNH₂NH₂ release, we determined the area of the corresponding HPLC peak and compared the value with that of 50 μM DoxNH₂NH₂. When treating DoxPt with 1 mM L-cysteine in pH

7.0 buffer at 37 °C, about 6% DoxNH₂NH₂ was released after 12 h (Fig. S26). In replicate experiments with 100 μM and 200 μM L-cysteine, no detectable amount of DoxNH₂NH₂ was observed after 12 h (Fig. S21-22).

If that is the case, what is the actual species that cause the observed cytotoxicity and the overcoming of resistance? The authors should try to find out what the newly formed species are as they may be responsible for the interesting properties described in the paper.

=> Many platinum agents, including clinically used oxaliplatin and cisplatin, react with L-cysteine and potentially other biomolecules, forming a complicated mixture of platinum-containing adducts. DoxPt also shares a similar feature, leading to promiscuous reactions with extracellular and intracellular biomolecules. NCI-60 analysis and RNA-i assay suggest that DoxPt kills cells via pathways similar to those of anthracyclines and classical platinum drugs. Further studies indicate that DoxPt overcomes efflux pump-mediated resistance by forming adducts with intracellular biomolecules, including various proteins. We tentatively attribute the observed activity of DoxPt to the combined effects of its adducts with such molecules. However, as shown by others (Ref. 17-33) and us, platinum compounds typically react with a wide range of biological nucleophiles to form a large number of adducts, especially in a complex cellular environment. Therefore, extensive work is required to identify each species and delineate its specific roles in bioactivity. Similar situations have been well-documented in the literature for structurally simple platinum agents used in the clinic. We thus believe that, although the suggested studies would provide valuable mechanistic insights, they are beyond the scope of the current work and are better addressed in future studies.

Other minor issues with the current response are described below.

-The authors write “RNAi signature assay showed that the modes of action of DoxPt resemble those of both doxorubicin and oxaliplatin” and in the main text they mention that they “compared the signature of DoxPt to that of Dox, Ox, and a physical mixture of Dox and Ox at equipotent doses.” However, only the DoxPt and physical mixture data are reported in Figure S90. The Ox and Dox data should also be reported for completeness. Also, some statistically significant differences can be observed for some shRNA and require a comment on why this is (or isn't) important in determining the overall mechanism of action of the DoxPt agent.

=> The data comparing DoxPt, Dox, Ox, and Dox-Ox mixture were reported in Table S3. We have re-plotted Fig. S90 (now Fig. S98) to show these results visually.

Comments on statistical significance have been included in the Supplementary Information.

-When reviewer 3 asks about differences in behaviour between the “released” DoxNH₂NH₂ and Dox in binding to major groove of DNA the authors responds that DoxNH₂NH₂ is, effectively, not released. However, this does not really answer the question about differences in behaviour compared to Dox. In fact, one could expect that, if DoxPt stayed intact, the binding to the minor groove of DNA would be even more affected, given that there would be not just a sugar but a full Pt complex attached to it. The authors should comment on this.

=> As the speciation of DoxPt in the complex cellular environment is not fully delineated, it is challenging to determine the exact species interacting with DNA to exert the killing effects. Even considering a simpler scenario that DoxPt targets DNA directly, it is unclear whether DoxPt (1) binds to nucleobases via Pt–nitrogen bonds, (2) binds in the minor groove with the anthracycline moiety, (3) solely acts as an intercalator, or (4) interacts by both covalent Pt–N bonds and non-covalent forces. Lacking sufficient structural information at the molecular level, we are reluctant to speculate on the DoxPt-DNA interactions in this manuscript. However, we appreciate this question and will conduct separate work investigating this topic in the future.